# *Col1a2*-Deleted Mice Have Defective Type I Collagen and Secondary Reactive Cardiac Fibrosis with Altered Hypertrophic Dynamics

**DOI:** 10.3390/cells12172174

**Published:** 2023-08-30

**Authors:** Stephanie L. K. Bowers, Qinghang Meng, Yasuhide Kuwabara, Jiuzhou Huo, Rachel Minerath, Allen J. York, Michelle A. Sargent, Vikram Prasad, Anthony J. Saviola, David Ceja Galindo, Kirk C. Hansen, Ronald J. Vagnozzi, Katherine E. Yutzey, Jeffery D. Molkentin

**Affiliations:** 1Division of Molecular Cardiovascular Biology, Cincinnati Children’s Hospital, University of Cincinnati, Cincinnati, OH 45229, USA; 2Center for Organoid and Regeneration Medicine, Greater Bay Area Institute of Precision Medicine (Guangzhou), School of Life Sciences, Fudan University, Guangzhou 511466, China; 3Department of Biochemistry and Molecular Genetics, University of Colorado Anschutz Medical Campus, Aurora, CO 80045, USA; 4Division of Cardiology, Consortium for Fibrosis Research and Translation, Department of Medicine, University of Colorado Anschutz Medical Campus, Aurora, CO 80045, USA

**Keywords:** type I collagen, cardiac hypertrophy, cardiomyocyte, fibroblast, extracellular matrix

## Abstract

Rationale: The adult cardiac extracellular matrix (ECM) is largely comprised of type I collagen. In addition to serving as the primary structural support component of the cardiac ECM, type I collagen also provides an organizational platform for other ECM proteins, matricellular proteins, and signaling components that impact cellular stress sensing in vivo. Objective: Here we investigated how the content and integrity of type I collagen affect cardiac structure function and response to injury. Methods and Results: We generated and characterized *Col1a2^−/−^* mice using standard gene targeting. *Col1a2^−/−^* mice were viable, although by young adulthood their hearts showed alterations in ECM mechanical properties, as well as an unanticipated activation of cardiac fibroblasts and induction of a progressive fibrotic response. This included augmented TGFβ activity, increases in fibroblast number, and progressive cardiac hypertrophy, with reduced functional performance by 9 months of age. *Col1a2-loxP*-targeted mice were also generated and crossed with the tamoxifen-inducible *Postn-MerCreMer* mice to delete the *Col1a2* gene in myofibroblasts with pressure overload injury. Interestingly, while germline *Col1a2*^−/−^ mice showed gradual pathologic hypertrophy and fibrosis with aging, the acute deletion of *Col1a2* from activated adult myofibroblasts showed a loss of total collagen deposition with acute cardiac injury and an acute reduction in pressure overload-induce cardiac hypertrophy. However, this reduction in hypertrophy due to myofibroblast-specific *Col1a2* deletion was lost after 2 and 6 weeks of pressure overload, as fibrotic deposition accumulated. Conclusions: Defective type I collagen in the heart alters the structural integrity of the ECM and leads to cardiomyopathy in adulthood, with fibroblast expansion, activation, and alternate fibrotic ECM deposition. However, acute inhibition of type I collagen production can have an anti-fibrotic and anti-hypertrophic effect.

## 1. Introduction

The extracellular matrix (ECM) is a dynamic protein-based network that provides structural support and serves as a platform for cellular functions and signaling in essentially all vertebrate tissues. A number of human diseases arise due to mutations in collagen I, a necessary core ECM protein, including osteogenesis imperfecta, chondrodysplasias, Alport syndrome, Caffey disease, atypical Marfan syndrome, and subtypes of Ehlers–Danlos syndrome [1]. Patients harboring these genetic mutations can present with a variety of symptoms, including cardiomyopathy. Excessive ECM deposition in the heart, generally referred to as cardiac fibrosis, is a common feature underlying a myriad of cardiac diseases [2]. Cardiac fibroblasts are the primary cell type regulating ECM composition and stiffness in the heart, in part by organizing collagen networks. With acute injury or chronic disease states, cardiac fibroblasts activate and transform to a contractile cell known as a myofibroblast, which expands and remodels the ECM network [3]. Defective collagen production by cardiac fibroblasts due to deletion of either heat shock protein 47 (HSP47) [4] or periostin [5] renders the heart unable to induce an effective fibrotic response, which secondarily impacts the ability of cardiomyocytes to hypertrophy with disease stimuli. Similarly, disrupting the activity of the transforming growth factor-β (TGFβ) receptors 1 and 2 in cardiac fibroblasts inhibited the disease-induced fibrotic response and reduced the ability of cardiomyocytes to hypertrophy [6,7]. These previous results suggest that activated fibroblasts are important in mediating cardiac hypertrophy in conjunction with their ability to augment ECM content.

ECM in the adult heart is primarily composed of type I collagen [8]. The triple-helix, rod-like type I collagen is formed by two α1(I) and one α2(I) procollagen chains encoded by the *Col1a1* and *Col1a2* genes, respectively [9,10]. The osteogenesis imperfecta mutant mouse (OIM), which has a recessive mutation in the *Col1a2* gene, models the human connective tissue disorder osteogenesis imperfecta. Due to the mutation, OIM/OIM homozygous mice have a severe reduction in functional collagen 1a2 [11]. As a compensatory response, collagen 1a1 chains can homotrimerize, although this generates collagen that is substantially weaker [11,12]. In the heart, collagen fibers from OIM/OIM mice were smaller in diameter and less dense [13]. The left ventricles of these mice were more compliant and readily expanded with the passive inflation of a balloon [13]. Hence, the homotrimeric collagen 1a1 fibrils that characterize the OIM mouse, while sufficient to support viability, are nonetheless structurally defective [12]. Indeed, the OIM/OIM mice have a high mortality rate following myocardial infarction (MI) due to rupturing of the myocardial wall [14].

Here, we used constitutive and inducible models of *Col1a2* deletion to investigate how the structure of type I collagen underlies ECM organization and content to support cardiac function, as well as how the expansion of type I collagen with disease/injury permits effective cardiac hypertrophy. We observed that developmental loss of a structurally rigorous type I collagen-containing ECM network in *Col1a2^−/−^* mice leads to a secondary reactive expansion of cardiac fibroblasts, enhanced total ECM deposition, increased transforming growth factor-β (TGFβ) activity, and cardiomyopathy with loss of cardiac function over time. However, by using an inducible *Col1a2-loxP* targeted allele we show that acute deletion of this gene from myofibroblasts inhibited new collagen production over 1 week of pressure overload that was associated with a reduction in the cardiac hypertrophic response. However, over several more weeks and the gradual production of defective collagen 1a1 homotrimers with additional fibrotic compensation, the full potential of the hypertrophic response was restored. Thus, the quality and quantity of type I collagen in the heart underlie structural ECM properties that cardiomyocytes and fibroblasts directly sense, both at baseline and in response to disease stimuli, impacting the hypertrophic potential of the heart.

## 2. Materials and Methods

### 2.1. Animals and Surgical Models

Mouse founders carrying the *Col1a2* knockout first allele (*Col1a2^−/−^*) were purchased from Mutant Mouse Resource & Research Centers (MMRRC, ID: 037695-UCD). To generate only *Col1a2 loxP*-targeted mice, the *Col1a2^−/−^* mouse was further crossed with *Rosa26-Flpe* females (B6.129S4-*Gt(ROSA)26Sor^tm1(FLP1)Dym^*/RainJ) to remove the neomycin cassette at the *frt* sites as described previously [15]. The *Postn-MerCreMer* (MCM) mice have also been previously described [15]. The *Rosa26 loxP* site-dependent reporter mice (*R26^NG−loxP−eGFP^*) were purchased from Jackson Laboratories (stock no. 012429) [16]. All crosses were carried out and maintained in the C57/BL6 strain background, and males and females were equally used in our studies. To induce Cre activity in *Postn^MCM^* mice in vivo, mice were fed tamoxifen-citrate chow (40 mg/kg body weight, Envigo-TD.130860) and/or were injected i.p. with tamoxifen (Millipore Sigma, T5648, St. Louis, MO, USA) that was dissolved in corn oil (75 mg/kg body weight/day). Transverse aortic constriction (TAC) surgery was performed to induce pressure overload and cardiac hypertrophy as described previously [17]. Briefly, a silk ligature was tied around a 26-gauge wire and the mouse transverse aorta to produce a pressure load on the heart. Myocardial infarction (MI) was induced by the permanent ligation of the left coronary artery as described previously [18]. After TAC, MI, or sham procedures, mice were treated with sustained-release buprenorphine (0.2 mg/kg) injected subcutaneously for pain management. Mice and incisions were monitored daily following surgery.

### 2.2. Animal Welfare and Ethics

Animals were handled in accordance with the principles and procedures of the Guide for the Care and Use of Laboratory Animals. All proposed procedures were approved by the Institutional Animal Care and Use Committee (IACUC) at Cincinnati Children’s Hospital (USA). Animal groups and experiments were handled in a blinded manner where possible. Randomization was not performed given that all mice were of the same genotypes and identical strain, and only age-matched littermates were compared. ARRIVE guidelines were followed in all mouse experimentation. No human materials or subjects were used. All mice were housed in a germ-free barrier environment with free access to food and water with 12 h day/12 h night cycles and observed every day by veterinary staff. Pain management in mice is discussed above.

### 2.3. Antibodies and Related Reagents

Antibodies against the following proteins were used: periostin (Novus Biologicals, Centennial, CO, USA; NBP1-30042; 1:300 dilution for IF, 1:1000 for Western blot); collagen I (Abcam, Boston, MA, USA; ab21286; 1:100 for IF); PDGFRα (R&D Systems, Minneapolis, MN, USA; AF1062; 1:100 for IF); collagen 1a2 (Santa Cruz, Dallas, TX, USA; sc-393573; 1:500 for Western blot); TGFb1 (Abcam 215715; 1:1000 for Western blot); TGFb2 (Abcam 205150; 1:1000 for Western blot); TGFb3 (Abcam 15537; 1:1000 for Western blot); pSmad2/3 (MaineHealth, Scarborough, ME, USA; D8591; 1:50 for IF); CD31 (BioLegend, San Diego, CA, USA; 102423; 1:100 for flow cytometry); CD45 (BD Biosciences, Franklin Lakes, NJ, USA; 563890; 1:100 for flow cytometry); MEFSK4 (Miltenyi Biotec, Waltham, MA, USA; 130-120-802; used 1:30 for flow cytometry). Wheat germ agglutinin (488) was from ThermoFisher, Waltham, MA, USA (W11261; 5 µg/mL for immunofluorescence).

### 2.4. Isolation of Cardiac Fibroblasts

Whole hearts isolated from anesthetized mice were briefly rinsed in cold sterile 1X PBS and the atria were removed. Whole ventricles were minced on ice using surgical scissors into approximately 2 mm pieces (8–10 pieces per mouse heart). Each dissociated ventricle was digested in 2 mL of DMEM containing 2 mg/mL of Worthington collagenase type IV (#LS004188), 1.2 U/mL dispase II (Roche, #10165859001, Basel, Switzerland), 0.9 mM CaCl_2_ and 2% fetal bovine serum (FBS) at 37 °C for 20 min with gentle rotation followed by manual trituration 12–15 times with a 10 mL serological pipette, such that all the tissue pieces were able to pass through the pipette. The tissues were then settled by sedimentation and the supernatant was passed through a 40 µm mesh strainer and stored on ice. Two milliliters of fresh digestion buffer were added, followed by 2 additional rounds of incubation, trituration, and replacement of supernatant with fresh digestion buffer, except trituration was performed with a 5 mL serological pipette for round 2 and a 1 mL p1000 pipette tip (USA Scientific, Ocala, FL, USA; #1112-1720) for round 3. The pooled supernatant from the 3 rounds of digestion was washed with sterile 1x PBS (adding 1x PBS to make a final volume of 40 mL) and centrifuged at 200× *g* for 20 min at 4 °C in a swinging bucket rotor centrifuge without brakes. The pellet was resuspended in Red Blood Cell Lysis buffer (155 mM NH_4_Cl; 12 mM NaHCO_3_; 0.1 mM EDTA) for 5 min at room temperature and centrifuged at 200× *g* for 20 min at 4 °C in a swinging bucket rotor centrifuge without brakes. The pellet was then resuspended in flow cytometry sorting buffer consisting of 1x HBSS (ThermoFisher Scientific, Waltham, MA, USA; Cat. #14025076) supplemented with 2% bovine growth serum (BGS) and 2 mM EDTA.

### 2.5. Flow Cytometry

Dissociated heart suspensions were prepared as described immediately above, and cell pellets were resuspended in 500 µL of FACS buffer (1x HBSS, 2% BGS, 2mM EDTA) in 1.5 mL LoBind Eppendorf tubes (ThermoFisher 13-698-791). Fluorophore-conjugated antibodies against CD31, CD45, and MEFSK4 were added (see above for antibody information and dilutions) and samples were incubated on a rotator at 4 °C for 30 min. Tubes were then centrifuged at 200× *g* for 8 min and pellets were resuspended in 500 µL fresh FACS buffer. Cell suspensions were filtered on ice into 5 mL polystyrene round-bottom tubes with a cell strainer cap (Falcon 352236). Samples were then analyzed using a BD FACSCanto cytometer running BD FACSDiVa V 8.0 software (BD Biosciences, Franklin Lakes, NJ, USA), with a violet (405 nm) laser to detect the Brilliant Violet 421-conjugated CD31 and CD45 antibodies and a red (633 nm) laser to detect the APC-conjugated MEFSK4 antibody. Initial gating with forward and side scatter was used to define single cells. Analysis and quantitation were defined as the total cell number of cells per milligram of dissociated tissue and were performed using FlowJo software (V10, Tree Star, Inc., Ashland, OR, USA).

### 2.6. RNA Extraction and Western Blots

Whole ventricles or isolated cells were digested in Trizol (ThermoFisher Scientific, Waltham, MA, USA; Cat. #15596018) for mRNA isolation. cDNA was synthesized using an iScript cDNA synthesis kit (Bio-Rad, Hercules, CA, USA) following the manufacturer’s protocol. Whole ventricles or isolated cells were digested in RIPA buffer with a cOmplete^TM^ Mini protease inhibitor cocktail (Millipore Sigma, Burlington, MA, USA; Cat. #4693124001) and PhosSTOP phosphatase inhibitor cocktail (Roche Diagnostics, Cat. #4906837001, Basel, Switzerland) for protein isolation. Ventricle ECM proteins were isolated using the Subcellular Protein Fractionation Kit for Tissues (Thermo Fisher Scientific, Waltham, MA, USA; Cat. #87790) following the manufacturer’s instructions. Equal amounts of protein were used for SDS gel electrophoresis and Western blots.

### 2.7. TGFβ Activity Assay

To determine changes in TGFβ activation, we used the TGF-beta 1 quantikine ELISA kit (R&D Systems, DB100C, Minneapolis, MN, USA). Briefly, 15–25 mg of heart tissue was crushed in 1x PBS + protease inhibitors (Millipore Sigma, Burlington, MA, USA; Cat.# 4693124001) using Biomasher^®^ tubes (FisherScientific, Waltham, MA, USA; Kimble 749625-0010), protein concentration quantified with a BCA Protein Assay Kit (Pierce, 23225, Appleton, WI, USA), and 200 µg of protein was used per well (samples were run in duplicate). All procedures for sample activation, standard curve preparation, and assay protocol followed the manufacturer’s instructions.

### 2.8. RNA Microarray

Whole mouse heart ventricles collected from either *Col1a2^+/−^* or *Col1a2^−/−^* at 2 months of age were lysed in Trizol for total RNA isolation followed by RNA quality assessment using an Agilent 2100 Bioanalyzer (Agilent Technologies, Santa Clara, CA, USA). Microarray analysis was performed using the Affymetrix Clariom S platform (Affymetrix, Santa Clara, CA, USA) at Cincinnati Children’s Hospital Medical Center Gene Expression Core Facility. Data in CHP files were analyzed using Transcriptome Analysis Console (Applied Biosystems, Foster City, CA, USA), Clariom_S_Mouse TAC Configuration file, and iPathwayGuide (Advaita Bioinformatics, Ann Arbor, MI, USA) to determine differential gene expression between experimental groups. The raw RNA microarray expression data were submitted to the GEO omnibus with an accession number of GSE204724 (embargoed until publication acceptance). Data are also simplified in Appendix A as significantly up- and downregulated transcripts between *Col1a2^+/−^* or *Col1a2^−/−^* hearts.

### 2.9. ECM Mass Spectrometry

Preparation of tissue samples was performed as previously described [19]. Briefly, 5 mg of lyophilized heart tissue samples were processed by a stepwise extraction with CHAPS and high salt, guanidine hydrochloride, and chemical digestion with hydroxylamine hydrochloride (HA) in Gnd-HCl generating cellular, soluble ECM (sECM), and insoluble ECM (iECM) fractions for each sample, respectively. The protein concentration of each fraction for each sample was measured using A660 Protein Assay (Pierce, ThermoFisher Scientific, Waltham, MA, USA). Thirty micrograms of protein resulting from each fraction was subjected to proteolytic digestion using a filter-aided sample preparation (FASP) protocol [20] with 10 kDa molecular weight cutoff filters (Sartorius Vivacon 500, Sartorius, Göttingen, Germany; #VN01H02). Samples were reduced with 5 mM tris(2-carboxyethylphosphine), alkylated with 50 mM 2-chloroacetamide, and digested overnight with trypsin (enzyme:substrate ratio: 1:100) at 37 °C. Peptides were recovered from the filter using successive washes with 0.2% formic acid. Aliquots containing 10 μg of digested peptides were cleaned using PierceTM C18 Spin Tips (ThermoFisher Scientific, Waltham, MA, USA; Cat. #84850) according to the manufacturer’s protocol, dried in a vacuum centrifuge, and resuspended in 0.1% formic acid in mass spectrometry-grade water.

Liquid chromatography-mass spectrometry (LC-MS/MS) was performed using an Easy nLC 1200 instrument coupled to an Orbitrap Fusion Lumos Tribrid mass spectrometer (all from ThermoFisher Scientific, Waltham, MA, USA), as previously described [19]. Fragmentation spectra were searched against the UniProt *Mus musculus* proteome database (Proteome ID # UP000000589 downloaded 1 December 2021) using the MSFragger-based FragPipe computational platform [21]. Contaminants and reverse decoys were added to the database automatically. The precursor-ion mass tolerance and fragment-ion mass tolerance were set to 10 ppm and 0.2 Da, respectively. Fixed modifications were set as carbamidomethyl (C), and variable modifications were set as oxidation (M), oxidation (P) (hydroxyproline), Gln->pyro- Glu (N-term), deamidated (NQ), and acetyl (Protein N-term). Two missed tryptic cleavages were allowed, and the protein-level false discovery rate (FDR) was ≤1%. The entire data set is shown in Appendix A.

### 2.10. Histology and Immunofluorescence Staining

Hearts were fixed in 4% paraformaldehyde (PFA) at 4 °C overnight. Tissues were then rinsed with 1x PBS and cryoprotected in 30% sucrose/1x PBS at 4 °C overnight before embedding in OCT (Tissue-Tek, Cat. #4583). Five-micron heart histological sections were collected and subjected to either picro sirius red staining or immunofluorescent staining [4]. Images of picro sirius red staining were acquired using an Olympus BX69 microscope with NIS Elements software (BR v. 5.42.02), and the fibrotic area in each image was determined using Image J (v1.50t, NIH free software). Images for immunofluorescent staining were acquired using an inverted Nikon A1R confocal microscope and quantified with NIS Elements AR 4.13 software.

### 2.11. Transmission Electron Microscopy

Hearts of anesthetized mice were perfused with 1% paraformaldehyde/2% glutaraldehyde (vol/vol) in cardioplegic solution (50 mmol/L KCl, 5% dextrose in 1x PBS), followed by 1% paraformaldehyde/2% glutaraldehyde (vol/vol) in 0.1 mol/L cacodylate buffer, pH 7.2. The heart was then removed and scars were isolated, divided into small fragments, and fixed in 1% paraformaldehyde/2% glutaraldehyde (vol/vol) in 0.1 mol/L cacodylate buffer, pH 7.2 at 4 °C, followed by post-fixation in 2% OsO_4_ (in 0.1 mol/L cacodylate buffer) before dehydration in acetone and embedding in epoxy resin. Ultrathin sections were counterstained with uranium and lead salts. Images were acquired on a Hitachi 7600 electron microscope equipped with an AMT digital camera.

### 2.12. Myocyte Contractility

Cardiomyocytes were isolated by Langendorff perfusion, as described previously [22]. Briefly, mice were injected with 100 U heparin and 10 min later hearts were excised, mounted to a cannula, and perfused with collagenase digestion buffer. Dissociated hearts were run through a 100-micron mesh filter and cardiomyocytes were allowed to settle for 10–15 min, rinsed, and calcium reintroduction was performed prior to taking measurements. Using an IonOptix data acquisition system for sarcomere detection, myocytes were paced at 10 V and 1 Hz; at least 8 myocytes were recorded per heart, and traces were analyzed using Ion Wizard software (IonOptix, Westwood, MA, USA; version 7.5.3.165).

### 2.13. Force Measurements on Decellularized Tissue Strips

To examine force generation of the myocardial ECM, the left ventricular free wall was cut into 3 mm (length) × 2 mm (width) strips using a slicing mold, dissecting scope, and ruler; 5–6 strips were obtained per heart. The strips were rinsed in 1x PBS and then digested for ~48 h in 1% SDS/1x PBS containing cOmplete^TM^ Mini protease inhibitor cocktail (Millipore Sigma, Burlington, MA, USA; Cat.# 4693124001) and PhosSTOP phosphatase inhibitor cocktail (Roche Diagnostics, Basel, Switzerland; Cat.# 4906837001). Since decellularization caused strips to lose some of their shape, strips were remeasured and trimmed to obtain 3 mm (length) × 2 mm (width) × 1 mm (height) for the experiment, and at least 4 strips were measured per heart. The decellularized tissue strips were attached to aluminum t-clips (Kem-Mil #1870) and mounted onto a muscle fiber test apparatus (Aurora Scientific, Aurora, ON, Canada; Model 802D-160-322) such that the initial tension of each strip was set to zero. Tissue length was increased by 5% over 50 ms, held for 150 ms, and then returned to baseline tension. This was repeated at 5% increments from 0 to 60% length increase for each strip. Force was monitored using DMC v600A software (Aurora Scientific). Change in force was calculated as the difference between the max force generated after the 50 ms pull and the minimum force achieved after each relaxation period. The minimum force was determined when the rate of force decay was zero and the relaxation rate was calculated by solving for the derivative of the best fit second-degree polynomial trend line. The slope of the derivative is the relaxation rate.

### 2.14. Cardiac Function by Echocardiography and Invasive Hemodynamics

Mice were anesthetized with 1.5% isoflurane and subjected to 2D guided M-mode echocardiography using a VisualSonics Vevo 3100 Imaging System (VisualSonics, Toronto, ON, Canada) with a 40 MHz transducer as described in [7]. Data were obtained by personnel blinded to genotype and treatment. For invasive hemodynamics, mice were anesthetized with 2.5% isoflurane. A high fidelity, solid-state 1.2F pressure-volume (PV) catheter (Transonic Scisense Inc., London, ON, Canada) was inserted into the left ventricle via right carotid cutdown and retrograde introduction of the PV catheter into the left ventricle. The signal was optimized by phase and magnitude channels [23]. Mice were normalized to 1.5% isoflurane and 37 °C for at least 5 min, at which point data were collected by PowerLab 8/36 and LabChart 7 Pro (both from ADInstruments, Colorado Springs, CO, USA). A minimum of 10 continuous seconds of recorded data were averaged for each time point.

### 2.15. Statistical Analyses

Data are expressed as mean ± SEM unless otherwise stated. mRNA and protein expression levels were normalized to GAPDH unless otherwise stated. The following tests were performed using GraphPad (Prism 9): Student’s *t*-test was performed for two group comparisons; a one-way or two-way ANOVA with Tukey’s post hoc analysis was performed for multiple group comparisons (specified in figure legends), as well as to determine the adjusted *p*-value of between-group comparisons. Statistical significance for each experiment is described in figure legends.

## 3. Results

### 3.1. Cardiomyopathic Phenotype in Germline Col1a2 Gene-Deleted Mice

To investigate the contribution of type I collagen to baseline heart function, we generated a mouse line carrying a *Col1a2* “knockout first allele” (see Section 2 and Figure 1A). The knockout first construction typically results in a null allele due to the insertion of a large coding cassette between the first coding exon and the rest of the gene. Indeed, RNA from whole ventricles of wild-type (*WT*) and *Col1a2^−/−^* mice showed a greater than 80% reduction of *Col1a2* expression by qRT-PCR (Figure 1B). As expected, however, there was no difference in immunostaining of total type I collagen in heart sections from *WT* versus *Col1a2^−/−^* mice at 3 or 6 months of age (Figure 1C,D). This observation is in agreement with previous findings, whereby loss of collagen1a2 in OIM mice is compensated by the incorporation of an additional collagen 1a1 chain as an obligate trimer [11,12]. One limitation of this mouse model is the placement of a large insertion cassette, which could result in unknown aberrant expression of neighboring genes. Our studies thus far seem to fall in line with the other models of *Col1a2* deficiency, but ultimately will need further validation in larger animal studies and human patients.

While both *Col1a2^+/−^* and *Col1a2^−/−^
*mice are viable and grossly normal, *Col1a2^−/−^
*mice uniquely developed progressive cardiac hypertrophy by 3 months of age. The ventricle weight to body weight ratio (VW/BW) of *Col1a2^−/−^
*mice increased more than 25% by 3 months of age compared to WT and *Col1a2^+/−^* littermates (Figure 1E), which remained at 9 months of age (Figure 1F). Interestingly, cardiac function was preserved until the mice reached 9 months of age, when a decrease in fractional shortening (Figure 1G) and dilation of the left ventricular chamber (Figure 1H) was observed. Using invasive hemodynamic measurements, we also observed an upwards and rightwards shift in the pressure-volume loops in the hearts of 10–12-week-old *Col1a2^−/−^
*mice compared to WT (Figure 1I), indicating increased afterload pressure with greater left ventricular distention. Importantly, at no point did we observe a functional difference between WT and *Col1a2^+/−^* mice, which were used as littermate controls. Each figure panel throughout the manuscript specifies which control line(s) were used for a given experimental dataset.

### 3.2. Col1a2^−/−^ Mice Have ECM Alterations and Fibrosis

Given the observations of cardiomyopathy, we used decellularized left ventricle myocardial tissue strips to assess the structural characteristics of the ECM in *Col1a2^−/−^
*hearts. As shown in Figure 2A, decellularized myocardial strips from *Col1a2^−/−^* mice failed to generate comparable force as WT strips with increased stretch. mRNA microarray analysis was also employed to interrogate the underlying molecular basis for cardiomyopathy in *Col1a2^−/−^* mice, which showed 225 upregulated and 249 downregulated genes in the heart due to the loss of *Col1a2* (Appendix A). Interestingly, the most prominent changes in gene expression were those related to a fibrotic response and/or genes that impact the mechanical properties of the ECM (Figure 2B). Gene Ontology (GO) enrichment analysis showed increased expression of a group of structure- and ECM-altering genes that included *Acta1*, *Postn*, *Fbn1*, *Loxl, Loxl2*, *Mmp19*, *Timp1*, and *Sparc* (labeled on the volcano plot in Figure 2B). Furthermore, we performed mass spectrometry of ECM from the left ventricles of hearts from *Col1a2^+/−^
*and *Col1a2^−/−^* mice. Consistent with the mRNA analysis, select ECM proteins were found to be changed in hearts from *Col1a2^−/−^* mice (Figure 2C and Appendix A), many of which are involved in the fibrotic process. One of the more interesting genes differentially expressed was that of *periostin*, which typically is induced to help mediate collagen maturation after injury [24]. Periostin deposition was also increased in *Col1a2^−/−^* adult hearts by immunostaining (Figure 2D). As fibroblasts are the main producers of periostin in the heart, we hypothesized that the cardiac fibroblasts might be altered in *Col1a2^−/−^* hearts. Using MEFSK4 as a fibroblast surface marker, FACS analysis showed an increase in the number of fibroblasts within *Col1a2^−/−^* hearts compared to heterozygous controls (Figure 2E,F). Indeed, analysis of platelet-derived growth factor receptor-α (PDGFRα) expression by immunohistochemistry showed a noticeable increase in hearts from *Col1a2^−/−^* mice versus the control (Figure 2G). Together, these findings indicate that in the absence of a rigorous collagen I network, cardiac fibroblasts expand and attempt to compensate for this structural weakness by inducing fibrotic gene expression.

### 3.3. Altered TGFβ Activity and Cardiomyocyte Function in Col1a2^−/−^ Hearts

TGFβ is a critical component of the fibrotic response and binds ECM proteins as a latent complex until activation. Both the microarray and mass spectrometry results indicated changes in TGFβ binding proteins in the hearts of *Col1a2^−/−^* mice (Figure 2B,C). Therefore, we investigated the content of TGFβ isoforms in the ECM-enriched portion of fractionated heart tissue samples via western blot. As shown in Figure 3A, reduced amounts of each of the three TGFβ isoforms were observed in the ECM of *Col1a2^−/−^* hearts compared to control littermates. Interestingly, however, these hearts showed an increase in TGFβ activity (Figure 3B), suggesting that the absence of TGFβ in the isolated ECM is due to its turnover and increased activity. Consistent with these data, we also detected an increase in phospho-Smad 2/3 immunostaining (nuclear and intracellular) in the hearts of *Col1a2^−/−^
*mice versus control (Figure 3C).

To investigate functional changes of cardiomyocytes induced by altered collagen organization and TGFβ activity, we used Langendorff-perfused hearts to isolate adult cardiac myocytes to assess contractility. Cardiomyocytes from *Col1a2^−/−^* hearts at 2–3 months (Figure 3D) and 8 months (Figure 3E) of age displayed significantly less contractility in response to electrical pacing. This corresponded with a significant increase in cardiomyocyte thickness, but not length (Figure 3F,G), further indicating secondary hypertrophy due to an increased reactive fibrotic program.

### 3.4. Col1a2^−/−^ Mice Have an Altered Hypertrophic Response to Pressure Overload

To investigate whether *Col1a2^−/−^
*mice could effectively respond to pressure overload hypertrophy, we performed transaortic constriction (TAC) when mice were 8 weeks old and harvested hearts after 7 days (timeline in Figure 4A). While *Col1a2* transcript increased in controls subjected to TAC, which remained absent in *Col1a2^−/−^
*hearts (Figure 4B), increased transcripts of *Col1a1*, *Postn*, *Col3a1,* and *Col5a1* were all detected in hearts of *Col1a2^−/−^* mice compared with heterozygous controls (Figure 4E,F), consistent with a greater fibrotic program due to the deletion of *Col1a2*. Importantly, even though germline *Col1a2^−/−^
*mice had larger hearts at baseline, these mice still effectively hypertrophied in response to TAC-induced pressure overload similar to heterozygous controls (Figure 4G). This latter observation suggests that sufficient compensatory fibrosis was in place in the hearts of germline *Col1a2^−/−^* mice to permit efficient cardiac hypertrophy. This interpretation is relevant because inhibition of the fibrotic response in *Hsp47* gene-deleted mice prevented pressure overload-induced fibrosis and attenuated the cardiac hypertrophic response [4].

### 3.5. Injury Responses Are Altered by Fibroblast-Specific Col1a2 Deletion in the Adult Heart

Given the baseline cardiomyopathy and augmented fibrosis observed in germline *Col1a2* gene-deleted mice, we then employed a conditional *Col1a2* deletion strategy to investigate the acute effects of fibrosis inhibition within the heart during disease stimulation. Accordingly, a tamoxifen-inducible *Postn-MerCreMer* gene-inserted mouse (*Postn^MCM^*) was crossed with a *Col1a2-loxP(fl)*-targeted mouse line to specifically delete this gene in myofibroblasts after injury (Figure 5A). Mice at 8 weeks of age were first subjected to myocardial infarction (MI) injury and fed tamoxifen chow 2 days post-surgery until tissue harvesting 2 weeks later (Figure 5B). The MI model was first employed given the overwhelming fibrotic response that must occur over 1 week’s time. Poor survival of *Postn^MCM/+^*; *Col1a2^fl/fl^
*mice, as was anticipated, only allowed us to investigate infarct/scar organization by transmission electron microscopy and not cardiac function. Importantly, scar regions at 2 weeks after MI injury showed abundant organized collagen fibrils in hearts from *Postn^MCM/+^* and *Col1a2^fl/f^* control groups, but not in *Postn^MCM/+^*; *Col1a2^fl/fl^
*experimental mice (all mice were given tamoxifen) (Figure 5C). These results indicate that acute deletion of *Col1a2* from myofibroblasts impedes the accumulation and organization of new type I collagen fibrils in the MI scar region of the heart. However, it is also possible that heterozygosity in the *Postn* locus due to the *MCM* cDNA insertion reduced total periostin expression, which might have impacted the effect of the *Col1a2* deletion phenotype.

Because of the high degree of lethality following MI in *Postn^MCM/+^*; *Col1a2^fl/fl^
*mice, pressure overload with TAC was employed to investigate their hypertrophic response (see Figure 5D for experimental timeline). Collagen I accumulation was inhibited in *Postn^MCM/+^*; *Col1a2^fl/fl^* mice after 1 week of TAC (Figure 5E), which was also associated with a significant reduction of cardiac hypertrophy (Figure 5G). However, by 2 weeks post-injury, and persisting until at least 6 weeks after, hypertrophy and fibrosis were similar between controls and *Postn^MCM/+^*; *Col1a2^fl/fl^
*mice (Figure 5F,G). These results indicate that in the absence of a sufficient fibrotic response to pressure overload stimulation, the heart hypertrophies less efficiently, but as the fibrotic response progresses and compensates for the loss of Col1a2, the hypertrophic response proceeds and can fully compensate.

## 4. Discussion

Here we used a mouse model whereby deletion of the *Col1a2* gene resulted in the absence of a structurally rigorous type I collagen-containing ECM network in the heart. Although these mice were viable, they developed overt cardiomyopathy by 9 months of age. At 3 months of age, hearts from these mice showed an expansion of cardiac fibroblast populations, induction of the fibrotic response, cardiomyocyte hypertrophy, and decreased myocyte contractility, all of which preceded the functional cardiac decline seen at 9 months. Furthermore, the disruption of the type I collagen network in *Col1a2^−/−^* hearts led to an increase in TGFβ activity, highlighting the importance of collagen I structural integrity to ensure proper growth factor signaling. These observations suggest that in the absence of structurally sound type I collagen in the heart, which directly effects TGFβ processing, a compensatory program is induced that creates a greater total fibrotic response in the heart that itself is likely disease-causing.

To study the effects of acute changes in fibrotic collagen deposition in the absence of baseline compensatory dysfunction, we used *Col1a2-loxP*-targeted mice and crossed them with *Postn^MCM^* mice to delete *Col1a2* specifically in activated myofibroblasts after injury. When *Col1a2* was acutely deleted from myofibroblasts in the adult heart, MI infarct scars were sparse and disorganized, and the overall survival of mice was poor presumably due to myocardial wall rupture. Using TAC to induce pressure overload, less cardiac hypertrophy was observed in *Postn^MCM/+^*; *Col1a2^fl/fl^
*mice after 1 week of stimulation, which was associated with less induction of new collagen production that normally occurs during this process [3]. Indeed, we previously showed that inhibition of fibroblast activation and their ability to generate new ECM in the heart reduced hypertrophic growth [4,5,6,7]. However, by 6 weeks of pressure overload, *Postn^MCM/+^*; *Col1a2^fl/fl^
*mice generated enough additional collagen, albeit defective, to restore hypertrophic potential. Overall, our results show that the structural integrity of type I collagen is sensed and responded to by cardiac fibroblasts and cardiomyocytes (and likely other myocardial cells), and that “fibrosis” can develop in multiple ways.

A remaining question is how exactly the content and organization of the ECM influence various cell populations present within the myocardium during cardiac developmental growth or disease-based remodeling. During postnatal development of the heart, cardiomyocyte growth is matched with transient cardiac fibroblast activation that reciprocally drives ECM production in a process that allows the heart to mature and generate more force [25]. Hence, it is likely that the cardiomyocyte directly senses the structural integrity of the ECM as it grows [26], which explains why acute inhibition of new ECM production results in less cardiac hypertrophy. In addition to ECM stiffness and structural support providing cardiomyocyte feedback, the ECM environment serves as a complex scaffold for latent growth factors that are released and activated during ECM remodeling, inflammation, or injury [27,28]. TGFβ is one such latent factor residing in the ECM, which when released by stretch or strain, programs fibroblast transformation to the myofibroblast, resulting in feedback expansion of the ECM during disease stimulation [29]. The epidermal growth factor family of latent growth factors are also released from the ECM where they can directly act on cardiomyocytes to program their hypertrophy, or by expansion of fibroblasts [30]. Mechanistically, a properly organized collagen network binds fibrillin and the latent TGFβ binding proteins (LTBPs) that together help control the proper latency and release of TGFβ [29]. Defective collagen would disrupt these complexes, possibly leading to constitutive TGFβ availability and ensuing fibrosis and disease, as well as the release of other latent growth factors. Our data suggest that collagen structure can affect TGFβ availability and activation, and suggest unknown mechanisms of action through TGFβ signaling outside of an acute injury. For example, the expression of LTBPs and other TGFβ binding proteins has not been fully studied with regard to their temporal, pathologic, or structural localization within the heart (reviewed in [31]). Defects in the structural rigor of the collagen network would also be directly sensed by cardiomyocytes through their integrin and dystroglycan-sarcoglycan attachment complexes to stimulate internal signal transduction that could also alter growth dynamics. Both mechanisms of affecting cardiac growth and remodeling are likely in play during acute disease stimulation and would be altered by defective type I collagen in *Col1a2^−/−^* mouse hearts.

## 5. Conclusions

Our results show that the quality and quantity of collagen in the heart affect cardiac fibroblast and cardiomyocyte function, likely through both direct mechanical sensing and indirect signaling mechanisms, such as underlying TGFβ activation. Fibroblasts respond to the developmental and acute deletion of *Col1a2* through a distinct fibrotic program, while cardiomyocytes concurrently alter their growth properties. Constitutive or inducible *Col1a2* deletion provides suitable models to study the molecular basis for human diseases that affect the organization of the basal cardiac type I collagen network.

## Figures and Tables

**Figure 1 cells-12-02174-f001:**
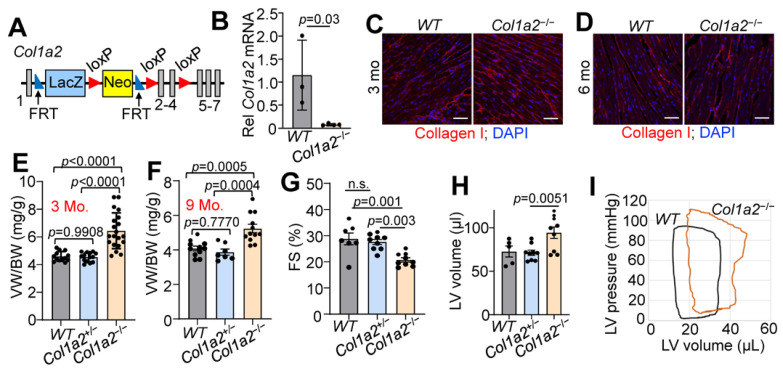
Germline deletion of *Col1a2* leads to cardiomyopathy in mice. (**A**) Schematic representation of the targeted *Col1a2* allele that was used for both constitutive and tissue-specific deletion of this gene in mice. Exons are numbered and shown as grey boxes, as are the positions of the FRT and LoxP sites. (**B**) Relative *Col1a2* mRNA expression in both wild-type (*WT*, *n* = 3) and null (*Col1a2^−/−^*, *n* = 4) mouse hearts at 3 months of age. (**C**,**D**) Representative immunohistochemistry from heart sections of type I collagen (red) staining intensity at 3 and 6 months of age in *WT* (*n* = 5) and *Col1a2^−/−^* (*n* = 4) mice with DAPI (blue) costaining to show nuclei. Scale bar: 25 μm. (**E**) Ventricle weight to body weight (VW/BW) ratio in *WT*, *Col1a2^+/−^
*and *Col1a2^−/−^
*mice at 3 months of age (*n* = 13, 14, and 20, respectively) and (**F**) 9 months of age (*n* = 11, 7, and 12, respectively). (**G**) Echocardiographic measurement of cardiac fractional shortening (FS%) in *WT* (*n* = 7), *Col1a2^+/−^
*(*n* = 9), and *Col1a2^−/−^
*(*n* = 9) mice at 9 months of age. (**H**) Echocardiographic measurement of left ventricular (LV) volume at diastole in *WT* (*n* = 5)*, Col1a2^+/−^
*(*n* = 9), and *Col1a2^−/−^
*(*n* = 9) mice at 9 months of age. (**I**) Representative baseline cardiac pressure-volume loops obtained by invasive hemodynamic measurements in hearts of *WT* and *Col1a2^−/−^* mice at 8–10 weeks of age. Data are presented as mean +/− standard deviation (SD). For (**B**,**D**), Student’s *t*-tests were used. For (**E**–**H**), one-way ANOVA analysis was used *p <* 0.05, and then Tukey’s post hoc analysis was used for between-group comparisons. Adjusted *p*-values are given in the figures.

**Figure 2 cells-12-02174-f002:**
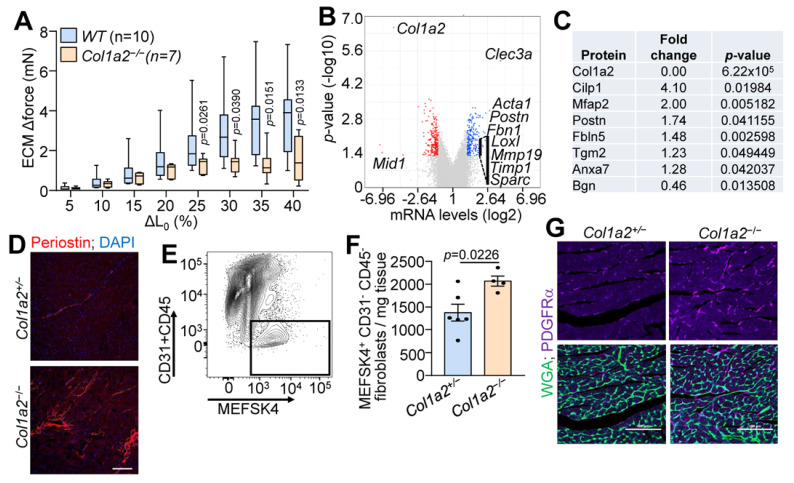
Hearts from *Col1a2^−/−^* mice show fibrobotic response. (**A**) Change in stretch-induced force generation of decellularized left ventricle ECM strips from *WT* and *Col1a2^−/−^* mice at 8–10 weeks of age. Data are presented as mean +/− SD. Multiple unpaired *t*-tests were used, *p*-values are stated in the figure. (**B**) Volcano plot of whole-ventricle mRNA microarray analysis of *Col1a2^−/−^* mouse hearts compared to *Col1a2^+/−^* at 2 months of age, *n* = 3 per genotype. (**C**) Mass spectrometry analysis of representative ECM proteins that were changed in *Col1a2^−/−^* mouse hearts compared to *Col1a2^+/−^
*hearts at 3 months of age, *n* = 4 per genotype. (**D**) Representative immunofluorescence images of periostin staining in heart histological sections from *Col1a2^+/−^* and *Col1a2^−/−^* mice at 3 months of age. Scale bar: 25 µm. (**E**) Flow cytometric gating strategy and (**F**) analysis of cardiac fibroblasts (MEFSK4^+^/CD31^−^/CD45^−^) from dissociated hearts of *Col1a2^+/−^* (*n* = 6) and *Col1a2^−/−^* (*n* = 4) mice at 3 months of age. Data are presented as mean +/− SD. Student’s *t*-tests were used and *p*-values are as stated. (**G**) Representative immunofluorescence images of platelet-derived growth factor receptor (PDGFR)-α (purple) staining from heart histological sections from *Col1a2^+/−^* and *Col1a2^−/−^* mice at 3 months of age, marking cardiac fibroblasts (purple), and wheat germ agglutinin (WGA) in green to show cellular outlines. Scale bar: 100 µm.

**Figure 3 cells-12-02174-f003:**
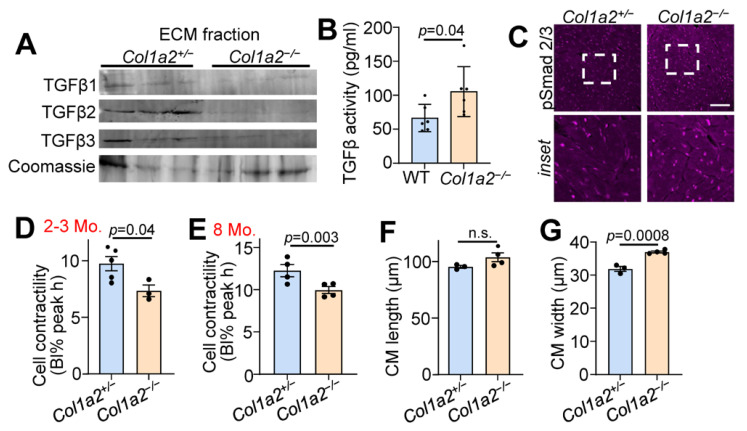
Hearts from *Col1a2^−/−^* mice have increased TGFβ activity and decreased cardiomyocyte contractility. (**A**) Western blot analysis of TGFβ isoforms in ECM fractions of hearts of 6-month-old *Col1a2^+/−^* and *Col1a2^−/−^* mice. (**B**) TGFβ activity in 3-month-old *WT* (*n* = 6) and *Col1a2^−/−^* (*n* = 6) hearts measured by ELISA. (**C**) Representative images of immunofluorescent staining of phospho-Smad2/3 in heart histological sections from *Col1a2^+/−^* and *Col1a2^−/−^* mice at 3 months age. Scale bar: 25 µm. The dashed white box in the upper row show the regionsmagnified in the lower panels. (**D**) Langendorff-dissociated myocytes from *Col1a2^+/−^* and *Col1a2^−/−^* mice were assessed for cell contractility at 2–3 months and (**E**) 8 months of age (*n* = 4/group, via IonOptix system, as discussed in Section 2.12). (**F**) Dissociated cardiomyocytes were fixed, mounted, and imaged to determine average cell length and (**G**) width (*n* = 3 *Col1a2^+/−^ and* 4 *Col1a2^−/−^*). For (**B**,**D**–**G**), data are presented as mean +/− SD. Student *t*-test was used and *p*-values are stated in each panel.

**Figure 4 cells-12-02174-f004:**
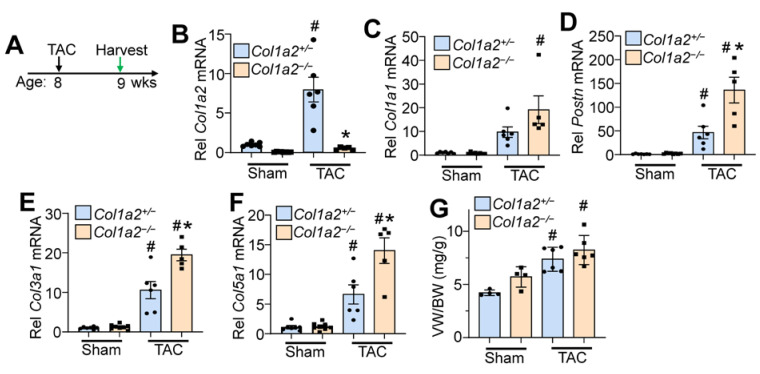
*Col1a2^−/−^* mice hypertrophy in response to pressure overload, despite a lack of *Col1a2*. (**A**) Schematic representation of the experimental design in *Col1a2^−/−^* mice that underwent TAC surgery at 8 weeks of age and tissue harvest 1 week later. Relative mRNA expression of (**B**) *Col1a2*, (**C**) *Col1a1*, (**D**) *Postn*, (**E**) *Col3a1*, and (**F**) *Col5a1* from hearts of *Col1a2^+/−^* and *Col1a2^−/−^* mice after 1 week of TAC. Animals per group: *n* = 7/8 for sham *Col1a2^+/−^* and *Col1a2^−/−^*, respectively; *n* = 6/5 for TAC *Col1a2^+/−^* and *Col1a2^−/−^*, respectively. (**G**) Measurements of ventricle weight to body weight ratio (VW/BW) in the indicated groups of mice (*n* = 4 in sham groups, *n* = 6 for TAC groups). For panels (**B**–**G**), all error bars are SD, and two-way ANOVA analysis *p <* 0.05, and Tukey’s post hoc analysis for between-groups comparison was performed; (**B**–**F**), ^#^
*p <* 0.001, compared to sham of same genotype; * *p <* 0.007, *Col1a2^+/−^* vs. *Col1a2^−/−^* TAC. For (**G**), ^#^
*p <* 0.03 compared to a sham of the same genotype.

**Figure 5 cells-12-02174-f005:**
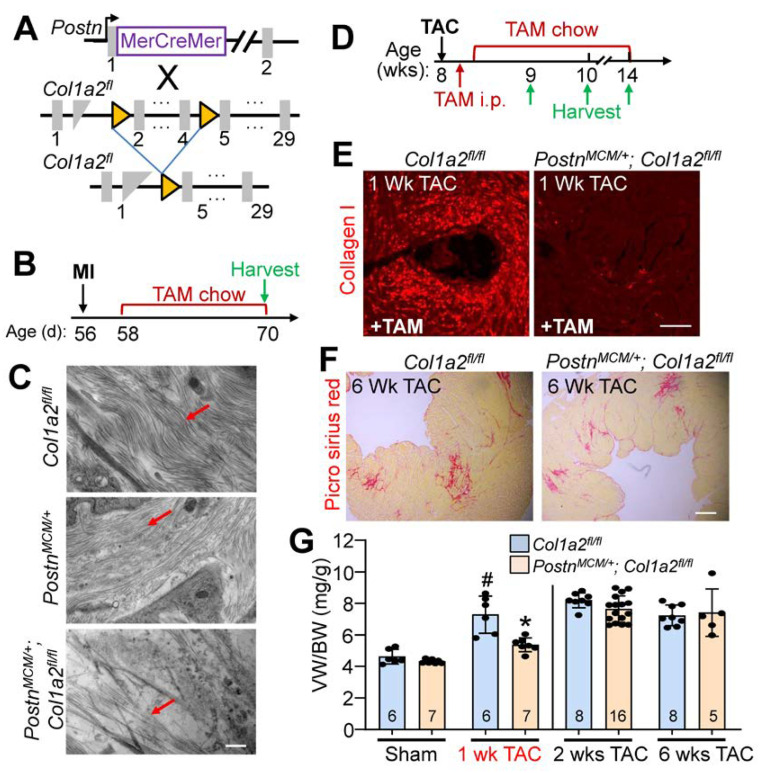
Fibroblast-specific *Col1a2* deletion in adult mice alters fibrosis and the hypertrophic response following injury. (**A**) Schematic of cell-specific Cre induction to ablate *Col1a2* in activated cardiac fibroblasts expressing the *Postn^MCM^* allele. (**B**) Experimental timeline with MI (myocardial infarction) and TAM (tamoxifen) chow feeding followed by harvest. (**C**) Transmission electron microscope images from infarcted myocardium 2 weeks after MI injury. Red arrows indicate collagen fibers in the forming scar region. Scale bar: 800 nm. (**D**) Timeline for TAC surgery, TAM administration/chow, and tissue harvest in *Postn^MCM/+^*; *Col1a2^fl/fl^
*and control mice starting at 8 weeks of age. (**E**) Representative immunohistological cardiac images of total type I collagen staining 1 week after TAC surgery in the 2 groups of mice shown. Scale bar: 25 μm. (**F**) Representative histological images of picrosirius red staining for fibrosis in control and *Postn^MCM/+^*; *Col1a2^fl/fl^
*hearts after 6 weeks of pressure overload. Scale bar: 100 μm. (**G**) Measurements of ventricle weight to body weight ratio following 1, 2, or 6 weeks of TAC in *Col1a2^fl/fl^
*and *Postn^MCM/+^*; *Col1a2^fl/fl^
*mice. For panel G, the number of mice per group is labeled on the graph and all error bars are SD. One-way ANOVA analysis *p <* 0.001, with Tukey’s post hoc analysis for between-group comparisons. ^#^
*p <* 0.001, compared to sham *Col1a2^+/−^*; * *p <* 0.001, *Col1a2^+/−^* vs. *Col1a2^−/−^* TAC.

## Data Availability

The raw RNA microarray expression data were submitted to the GEO omnibus with an accession number of GSE204724 (embargoed until publication acceptance). All other data are contained within the manuscript or the supplements.

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
