# Peer review of "Col1a2-Deleted Mice Have Defective Type I Collagen and Secondary Reactive Cardiac Fibrosis with Altered Hypertrophic Dynamics"

_cells, 2023, doi:10.3390/cells12172174_

Round 1

Reviewer 1 Report

The manuscript of Bowers et al. (2023) examines the developmental versus acute post-natal cardiac role of type 1 collagen (Col1a2).  The manuscript is clearly written, provides definitive data regarding developmental versus post-natal biologic impact of Col1a2, making use of state of the art methodologies including germline and inducible gene targeting models, gene expression analysis, proteomic assessments along with a wide array of physiologic and morphologic measures.  The authors make a convincing argument that developmental Col1a2 expression has emergent consequences for cardiac adaptation as animals age, which is quite distinct from the acute role of Col1a2 in stabilizing ECM in response to pressure overload or in the more severe challenge of myocardial infarction.  The compensatory fibrotic response in the absence of Col1a2 is also quite interesting, and suggests that ECM/fibrosis gene expression is governed by a sensitive rheostat mechanism than initially understood.  Together, the data in this manuscript provides for a critical clarification in the field, highlighting the dichotomy in collagen affects on cardiac biology.  However, prior to publication, the authors should address a number of minor concerns noted below.    

Minor concerns 

  1. The authors state that wildtype and Col1a2+/- are used interchangeably as controls.  Wildtype and heterozygote models are often considered phenotypic equivalents in the literature. However, the authors should specify exactly which experiments this applies to, as it is unclear from the manuscript itself.  
  2. The authors state that total collagen does not change in younger Col1a2-/- mice versus wildtype.  However, this comparison time point (3 months of age) precedes the emergence of the phenotypic cardiac differences that emerge in older Col1a2-/- animals.  Indeed, the authors are assuming that there is a compensatory inclusion of Col1a1, that may or may not change in older animals.  At a minimum, the authors should include gene expression via qPCR of Col1a1 in younger versus older animals to address this issue (the microarray data in 2 month old animals is unlikely to be definitive for age related alterations in collagen transcripts).  Indeed a wide cohort of profibrotic markers do increase in the the Col1a2-/- hearts which itself suggests there may be a compensatory upregulation in collagen 1 at later stages. Perhaps the authors can also query their mass spec data, was Col1a1 protein detactable in the MS data? 
  3. The authors should provide clear immunofluorescence images in Figure 3C, as it is difficult to discern differences in pSMAD staining in wildtype versus gene targeted conditions. 
  4. With regards to the acute assessment experiment of Col1a2 deletion, why did the authors start the inducible chow 2 days post surgery?  Does this dovetail with the animals ability or desire to eat?  I would anticipate that an earlier induction of the KO, via the inducible chow mix, at 1 day post surgery, may have a different and/or more sever outcome that would persist beyond the early time point sampled in Figure 5. 

Reviewer 2 Report

In their manuscript titled “Col1a2-deleted mice have defective type I collagen and secondary reactive cardiac fibrosis with altered hypertrophic dynamics”, Bowers et al investigate the effect of Col1a2 knockout on cardiac remodeling in mouse models of cardiac fibrosis. Collagen is the backbone of scar tissue.  Collagen type 1 is the predominant form of collagen in scar tissue regardless of species or organ. Collagen fibers that form the scar tissue are composed of triple helix with one Collagen type 1 alpha 2 and two collagen type 1 alpha 1 units. Knock-down of Collagen type 1 alpha 2 leads to compensation by formation of homotrimers that are weaker in structure than the heterotrimer. In this manuscript, the authors investigated how the content and integrity of type I collagen affects cardiac structure-function and response to injury. The authors generated two Col1a2  knockout models, one constitutive and one conditional. 

In the germline deletion of Col1a2 in mice, the authors first confirmed knockout of Col1a2, by mRNA analysis. Then the authors showed through Collagen I staining that the amount of collagen is unaltered in the knockout mice compared to WT. However, the ventricle weight at 3 and 9 months was higher in the KO model compared to the heterozygous or WT mice. This was accompanied by a lower Fractional shortening, an increased LV volume and an altered pressure/volume loop in knockout mice.  The authors next looked at the tissue and fibrosis response in these mice and found that the ECM in knock out mice was weaker than in WT mice, failing to generate comparable force upon stretching. Additionally, mRNA analysis showed increased expression of a group of structure- and ECM-altering genes including  Acta1PostnFbn1Loxl, Loxl2Mmp19Timp1 and Sparc. Protein analysis by MS showed that  some ECM proteins were are changed in hearts from Col1a2-/- mice, including periostin. Perionstin was actually increased in Col1a2-/- mice  and a corresponding increase in the number of fibroblasts within Col1a2-/- hearts compared to heterozygous controls. The authors conclude that these findings indicate that in the absence of a rigorous collagen I network, cardiac fibroblasts expand and attempt to compensate for these structural weaknesses by inducing fibrotic  gene expression. The authors next investigated the content of TGFβ isoforms in the ECM of heart tissue samples via western blot and found reduced amounts of each of the three TGFβ isoforms was observed in the ECM of Col1a2-/- hearts compared to control littermates  but these hearts showed an increase in overall TGFβ activity. The authors also found that cardiomyocytes from Col1a2-/- hearts at 2-3  and 8 months of age displayed significantly less contractility and increase in cardiomyocyte thickness.  The authors then performed transaortic constriction on 8 weeks old Col1a2-/-  mice and harvested hearts after 7 days. Col1a2 transcription increased in control TAC mice but remained  absent in Col1a2-/- hearts.  Transcription of Col1a1PostnCol3a1 and Col5a1 were all higher in hearts of Col1a2-/- mice compared with heterozygous controls.

Finally, the authors performed similar investigation in conditional Col1a1 knockout mice. The authors found by TEM  that scar regions at 2 weeks after injury showed organized collagen fibrils in hearts from Post- 440 nMCM/+ and Col1a2fl/f control groups, but not in PostnMCM/+; Col1a2fl/fl experimental mice. These results indicate that acute deletion of Col1a2  from myofibroblasts impedes the accumulation and organization of new type I collagen  fibrils in the MI scar region of the heart. The authors also used TAC pressure overload model in these acute deletion mice and found that total collagen I accumulation was inhibited in PostnMCM/+;Col1a2fl/fl mice after 1 week of TAC   which was reflected in the significant reduction of cardiac hypertrophy after 1 week of TAC . However, at week 2  until week 6 after injury hypertrophy and fibrosis were similar in the two groups. 

The work in this manuscript by Bowers et al is overall impressive and substantial. The writing and figures are clear and easy to read and understand.  However, there are a few issues that require attention and that should be addressed prior to publication. These are mostly regarding the validation of the second model that was used in this study.  The conditional, tamoxifen-induced deletion of Col1a2 was not verified. Additionally, investigation of  PostnMCM/+447 Col1a2fl/fl effect on base-line collagen compared to wild type or control was not completed. The authors did a great job with this investigation in the germline knockout mice, but failed to do the same rigorous investigation in this second knockout model. These analyses must be completed in order to accurately  interpret the results presented in Figure 5. 

A few other points:

1)     It is not clear why MI model was not performed in the germline deletion of Col1a2

2)     Figure 5C: both postnMCM/+ and postnMCM/+;Col1a2fl/fl appear to have similar level of  collagen organization that is different from the Col1a2fl/fl group. It seems that postnMCM/+ itself influences collagen organization. Please comment. Is the organization quantifiable?  

3)     Why is Total collagen I accumulation inhibited in PostnMCM/+447 Col1a2fl/fl mice after 1 week of TAC? Does this occur at baseline also(without TAC)? This point needs clarification

Reviewer 3 Report

In the current manuscript, the authors described that Col1a2 is essential to maintain the structure of cardiac ECM fiber using Col1a2 knockout mouse models. The discovery is an important supplement to current knowledge about the dysfunction of Col1a2. Here are some points that could help to improve the current manuscript:

-One major concern for this manuscript is the use of the tm1a mouse model as the global knockout mouse model to perform most of the experiments. In this mouse model, a huge fragment containing LacZ and Neo Cassatt was inserted into the Col1a2 locus. Although Col1a2 is knocked out by this strategy, huge insertion would lead to some unknown effects using these mice. I would recommend adding this point as a limitation of the current study in the discussion section.

-Just curious, is there any difference in lifespan in Col1a2 knockout mice compared to control, as the ECM fiber strength is much weaker in knockout mice?

-In most of the figures, the N number is missing in the figure legends. In the mice work in this manuscript, are both male and female mice used in all the experiments? This point should be clarified in the figure legends and methods section.

-It’s weird to observe that all TGFb proteins are decreased while the activity of TGFb is enhanced in Figure 3. It’s hard to believe that this is due to the turnover of TGFb. What’s the expression change of TGFb turnover genes in the knockout hearts? Is there any change in immune cell infiltration in the knockout hearts? This inconsistency in the expression and activity of TGFb should be discussed more thoroughly.

-For the TAC surgery in the global knockout model in Figure 4, what’s the change of cardiac function in these conditions? BTW, in Figure 4, two-way ANOVA is more suitable for the statistical evaluation of the results with two variants.

-Some minor issues: Table S2 is missing; TCFMCM mouse is not used in this manuscript while it is described in the material methods part; For RNA microarray, sometimes it was described as RNA sequencing. Please correct it in the manuscript.

Round 2

Reviewer 3 Report

The authors have addressed all my concerns. It's suitable to publish in current version.